# Rapid Diagnostic Tests to Guide Case Management of and Improve Antibiotic Stewardship for Pediatric Acute Respiratory Illnesses in Resource-Constrained Settings: a Prospective Cohort Study in Southwestern Uganda

Emily J. Ciccone,[a] Lydia Kabugho,[b] Emmanuel Baguma,[b] Rabbison Muhindo,[b] Jonathan J. Juliano,[a] Edgar Mulogo,[b] Ross M. Boyce[a,b,c]

[a]Division of Infectious Diseases, University of North Carolina School of Medicine, Chapel Hill, North Carolina, USA
[b]Department of Community Health, Mbarara University of Science and Technology, Mbarara, Uganda
[c]Department of Epidemiology, Gillings School of Global Public Health, Chapel Hill, North Carolina, USA

**ABSTRACT** Pediatric acute respiratory illness (ARI) is one of the most common reasons for evaluation at peripheral health centers in sub-Saharan Africa and is frequently managed based on clinical syndrome alone. Although most ARI episodes are likely caused by self-limited viral infections, the majority are treated with antibiotics. This overuse contributes to the development of antimicrobial resistance. To evaluate the preliminary feasibility and potential impact of adding pathogen-specific and clinical biomarker diagnostic testing to existing clinical management algorithms, we conducted a prospective, observational cohort study of 225 children presenting with malaria-negative, febrile ARI to the outpatient department of a semiurban peripheral health facility in southwestern Uganda from October 2019 to January 2020. In addition to routine clinical evaluation, we performed influenza and *Streptococcus pneumoniae* antigen testing and measured levels of C-reactive protein, procalcitonin, and lactate in the clinic's laboratory, and conducted a follow-up assessment by phone 7 days later. Almost one-fifth of participants (40/225) tested positive for influenza. Clinical biomarker measurements were low with C-reactive protein of >40 mg/L in only 11% (13/222) of participants and procalcitonin >0.25 ng/mL in only 13% (16/125). All but two children received antibiotic treatment; only 3% (7/225) were admitted. At follow-up, 59% (118/201) of caregivers reported at least one persistent symptom, but fever had resolved for all children. Positive influenza testing was associated with persistent symptoms. In summary, we demonstrate that simple, rapid pathogen-specific testing and biomarker measurement are possible in resource-limited settings and could improve syndromic management and, in turn, antibiotic stewardship.

**IMPORTANCE** Globally, respiratory illness is one of the most common reasons that children seek care. It is often treated inappropriately with antibiotics, which can drive the development of antibiotic resistance. In resource-rich settings, testing for specific pathogens or measurement of clinical biomarkers, such as procalcitonin and C-reactive protein, is often employed to help determine which children should receive antibiotics. However, there are limited data on the use of these tests in resource-constrained, outpatient contexts in sub-Saharan Africa. We enrolled children with respiratory illness presenting to a clinic in southwestern Uganda and performed testing for influenza, *Streptococcus pneumoniae*, C-reactive protein, and procalcitonin on-site. Almost all children received antibiotics. We demonstrate that employing clinical algorithms that include influenza and clinical biomarker testing could significantly decrease antibiotic prescriptions. Our study therefore provides preliminary data to support the feasibility and potential utility of diagnostics to improve management of respiratory illness in resource-constrained settings.

**KEYWORDS** antimicrobial stewardship, pediatric infectious disease, Uganda, biomarkers, influenza, diagnostics, respiratory infections

Address correspondence to Emily J. Ciccone, ciccone@med.unc.edu.

Acute respiratory illness (ARI) is the second leading cause of mortality worldwide in children under 5 years of age (1) and one of the most common reasons for care-seeking in sub-Saharan Africa (SSA). With the widespread uptake of rapid, point-of-care diagnostic tests that allow for parasitological confirmation of malaria infection (i.e., malaria rapid antigen testing [mRDTs]), it is increasingly apparent that malaria accounts for only a small subset of febrile illness visits (2, 3). Similar tools to discriminate causes of ARI when the mRDT is negative are not routinely available. The preponderance of these children will have self-limited, upper respiratory tract infections, likely of viral etiology (4–6). However, a small proportion will have bacterial pneumonia. While the fraction of children presenting with febrile illness who ultimately require antibiotics or hospital admission is small, the substantial morbidity and mortality associated with bacterial pneumonia mean that timely and accurate diagnosis and treatment are critical.

Accurately identifying the subset of children with bacterial pneumonia in resource-limited settings, where diagnostic imaging (e.g., chest radiography) and robust laboratory infrastructure are not routinely available, is challenging. Furthermore, outpatient providers at peripheral health centers, where most patients first seek care, are generally nonphysicians with limited clinical training. In these circumstances, diagnosis and management of ARI is based on the presence or absence of clinical symptoms and physical findings. This approach is supported by the World Health Organization's (WHO) Integrated Management of Childhood Illnesses (IMCI) guidelines, which recommend antibiotic treatment based only on symptoms of cough and fast breathing (7). While there is no definitive gold standard for diagnosing bacterial pneumonia, previous studies have established that clinical symptoms are not specific to a causative organism (8). In addition, respiratory rate, a key component of the evaluation for pneumonia, can be difficult to measure precisely and consistently (9, 10). This dependence on syndromic diagnosis likely leads to both (i) an overuse of antibiotics, which can cause adverse effects and drive antimicrobial resistance, and (ii) an underrecognition of children at high risk for bacterial infection (11–13).

Therefore, the development and validation of a method to distinguish patients in whom antibiotic treatment can be safely avoided from those requiring expedited treatment and/or referral is an important step toward reducing child mortality and improving antibiotic stewardship in low-resource settings. Point-of-care (POC) measurement of biomarkers, such as C-reactive protein (CRP) and procalcitonin (PCT), and pathogen-specific testing may help identify patients in whom antibiotics are not needed (14–17). While frequently employed in resource-rich settings, there is limited use of available point-of-care tests for clinical biomarkers in SSA and in Uganda in particular. To evaluate the potential impact of adding pathogen-specific assays and more extensive clinical biomarker testing to existing clinical management algorithms, we conducted a prospective, observational cohort study of children presenting to the outpatient department (OPD) of a peripheral health facility in southwestern Uganda.

## RESULTS

From October 2019 to January 2020, 309 children between 1 and 10 years of age presenting with fever and at least one respiratory symptom to the OPD at Kasese Health Center in southwestern Uganda were screened for eligibility. Of those, 13% (41/309) were mRDT-positive, while the caregivers of 43 children either elected not to participate or were not present to provide consent (Fig. 1). Demographic and clinical characteristics of the 225 enrolled participants are shown in Table 1. Most children were under 5 years of age; the median age was 3 years (interquartile range [IQR], 2–5). The majority were from areas near Kasese town, and the proportion who lived in houses with permanent roofs (e.g., iron sheets; associated with higher socioeconomic status) was similar to that which was reported in the most recent National Population and Housing Census profile of Kasese District (18). Only one child had been seen previously for the same illness, and none had received treatment prior to presentation. One child was known to be HIV-infected at the time of enrollment and was receiving antiretroviral therapy; two additional children tested

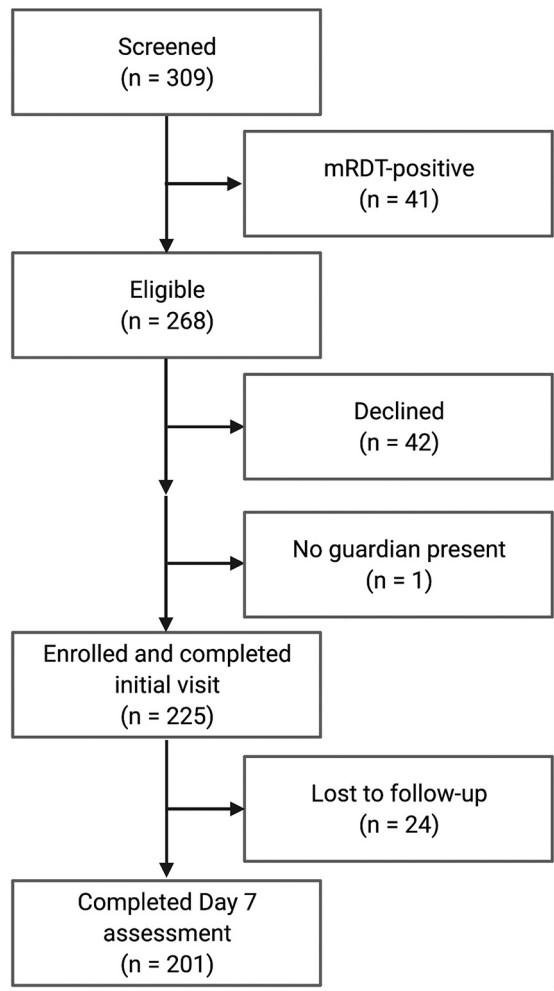

**FIG 1** Flow diagram of screening and enrollment process for the prospective, observational cohort study of children presenting with febrile acute respiratory illness to Kasese Health Center from October 2019 to January 2020.

positive for HIV by rapid testing during the initial visit. None of the participants reported other known chronic medical conditions such as asthma, diabetes, epilepsy, or heart problems.

**Clinical presentation and management.** The participants' clinical presentations are summarized in Tables 1 and 2. The most common reported symptoms were fever, cough, and rhinorrhea. The median duration of fever was 5 days (IQR, 5–5). Danger signs, including chest in-drawing, coma, and convulsions, were uncommon. Although fast breathing was infrequently reported by caregivers, almost half of the children in whom the respiratory rate (RR) was measured were tachypneic for age. None of those who were tachypneic by clinician measurement also self-reported chest in-drawing. A total of seven children (3%) were admitted to the inpatient department, four of whom were classified as moderately or severely underweight per WHO weight-for-age standards (19). None of the children with hypoxia were admitted. Of the 225 participants, 134 had a RR documented by the clinician. Of those children, 69 (51%) did not meet criteria for diagnosis of pneumonia per Uganda Ministry of Health clinical guidelines (20). Including all 225 children regardless of whether RR was documented, 148 (66%) did not meet diagnostic criteria for pneumonia per local guidelines.

All but two children (223/225, 99%) received antibiotic treatment, including all of the children with RR documented who did not meet clinical criteria for pneumonia. The most commonly prescribed antibiotics were amoxicillin (77%) and cotrimoxazole (11%), which are the first- and second-line treatments, respectively, for pneumonia. Most children who

**TABLE 1** Demographic and clinical characteristics at initial visit for participating children with malaria-negative febrile respiratory illness presenting to the outpatient department at Kasese Health Centre from October 2019 to and January 2020 ($n$ = 225)

| Demographic characteristic | *n* (%) |
|---|---|
| Age in yrs (median, IQR) | 3 (2–5) |
| 1-5 yrs | 161 (72) |
| ≥5 yrs | 64 (28) |
| Sex | |
| Female | 129 (57) |
| Male | 96 (43) |
| Home location | |
| Urban (Kasese town) | 5 (2) |
| Peri-urban | 210 (93) |
| Rural (village) | 10 (4) |
| Family size (median, IQR) | 5 (3–6) |
| Home construction | |
| Brick with iron sheets | 148 (66) |
| Mud with iron sheets | 71 (32) |
| Concrete | 3 (1) |
| Mud with grass thatched roof | 3 (1) |
| Guardian occupation | |
| Subsistence farmer | 112 (50) |
| Businesswoman | 77 (34) |
| Health worker | 14 (6) |
| Teacher | 9 (4) |
| Tailor | 5 (3) |
| Other | 8 (4) |
| Owns bed net | 183 (84)[a] |
| **Clinical presentation** | ***n* (%)** |
| Days of fever (median, IQR) | 5 (5–5) |
| Previously seen for same condition | 1 (0.4)[b] |
| Antibiotic treatment in the last 2 wks | 0 (0) |
| Antimalarial treatment in the last 2 wks | 0 (0) |
| Abnormal vital signs | |
| Fever (temp ≥38°C) | 34 (15) |
| Hypoxia (SpO2 < 90%) | 7 (3) |
| Tachypnea | 63 (47)[c] |
| Weight-for-age z-score, < −2.0[b,d] | 45 (20) |

[a]Data available for 219 of the 225 participants.
[b]Data available for 224 of the 225 participants.
[c]RR measured in 134 of the 225 participants.
[d]Moderately (weight-for-age < −2.0 and ≥ −3.0 SD of the median) or severely underweight (SD of the median < −3.0) per WHO guidelines (19).

were prescribed cotrimoxazole (20/24, 83%) were evaluated in January 2020. Children who received cotrimoxazole were slightly older than those who received amoxicillin, although the difference was not statistically significant (median age, 4.5 versus 3 years; $P$ = 0.086). Six of the seven children admitted to the inpatient ward received parenteral ceftriaxone. Nine children (4%) were prescribed more than one antibiotic.

**Pathogen-specific testing.** All participants underwent testing for influenza, and 18% (40/225) were positive (Fig. 2). Over the entire study period, positive tests were evenly distributed between subtypes A (21/40, 53%) and B (19/40, 48%), with both subtypes circulating in the community at the same time. However, the relative proportion of influenza A and B differed by month, with the majority of cases being subtype B in December and subtype A in October and January. We did not identify an association between age, sex, home location, or home construction material and influenza test result (data not shown). All participants who tested positive for influenza of either subtype received antibiotics.

Of the 214 children who were able to provide a urine sample for *Streptococcus pneumoniae* urinary antigen (SpUA) testing, 31% (66/214) were positive. None of those who tested positive also had a CRP of >80 mg/L, the combination of which has been shown to be predictive of pneumococcal pneumonia (21).

**TABLE 2** Symptoms reported by caregivers of participating children with malaria-negative febrile ARI at initial presentation (n = 225) and follow-up assessment after seven days (n = 201)

| Symptom | Initial presentation n (%) | Follow-up assessment n (%) |
|---|---|---|
| Fever | 213 (95) | 0 (0) |
| Fast breathing | 19 (8) | 3 (1) |
| Cough | 220 (98) | 100 (50) |
| Wheezing[a] | 4 (2) | 0 (0) |
| Chest in-drawing[b] | 1 (0.4) | 2 (1) |
| Rhinorrhea | 216 (96) | 86 (43) |
| Diarrhea | 16 (7) | 0 (0) |
| Vomiting | 19 (8) | 1 (0.5) |
| Anorexia/poor feeding | 32 (14) | 23 (11) |
| Convulsions[b] | 2 (1) | 0 (0) |
| Coma[b] | 0 (0) | 0 (0) |
| Headache | 26 (12) | 5 (2) |

[a]Not asked about specifically during symptom assessment but reported as "other" symptom.
[b]Danger sign per Uganda Ministry of Health Clinical Guidelines (20).

**Biomarker testing.** The CRP, PCT, and lactate testing results are detailed in Table 3. Overall, the measurements were low for most children; the median CRP was 8 mg/L (IQR, 8 to 12 mg/L), PCT was 0.1 ng/mL (IQR, 0.1 to 0.11 ng/mL), and lactate was 1.9 mmol/L (IQR, 1.1 to 2.8 mmol/L). CRP and PCT were positively associated ($\rho = 0.50$, $P < 0.001$), but neither was associated with lactate (CRP $\rho = 0.10$, $P = 0.17$; PCT $\rho = 0.09$, $P = 0.34$). When employing the CRP and PCT cutoffs of 40 mg/L and 0.25 ng/mL, respectively, used in previous studies (22, 23), we noted moderate concordance ($\kappa = 0.50$, $P < 0.001$) between the two biomarkers, but poor concordance between a lactate of >3.5 mmol/liter and either CRP ($\kappa = 0.06$, $P = 0.19$) or PCT ($\kappa = 0.05$; $P = 0.30$). None of the three biomarkers differed between those who met the clinical definition for pneumonia per national guidelines and those who did not. PCT, but not CRP, was different between influenza-positive and influenza-negative individuals ($P = 0.03$), but median values were the same for both groups, with overlapping confidence intervals (CI) (influenza-negative, 0.1 [95% CI 0.1 to 0.1]; influenza-positive, 0.1 [0.1 to 0.18]).

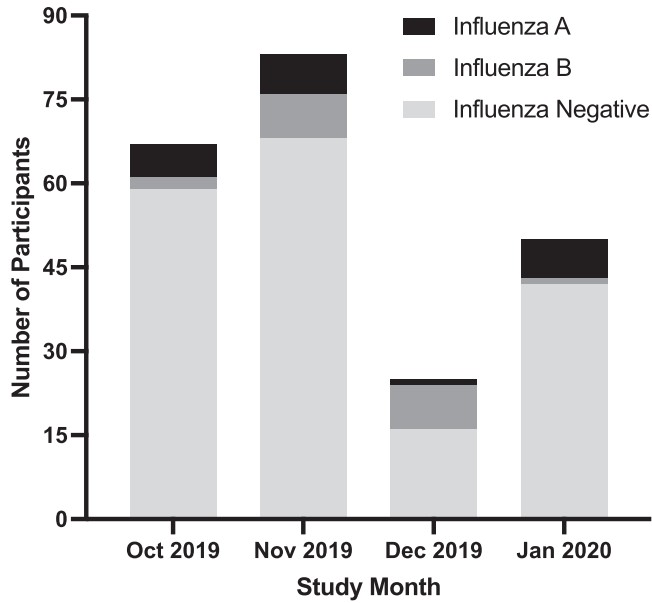

**FIG 2** The number of participating children with malaria-negative febrile ARI presenting to the outpatient department at Kasese Health Center from October 2019 to January 2020 who had each of the possible influenza test results by study month.

**TABLE 3** Clinical biomarker measurements among participating children with malaria-negative febrile ARI presenting to the outpatient department at Kasese Health Center from October 2019 to January 2020

| Test | n (%) | Median (IQR) |
| --- | --- | --- |
| CRP (mg/L; n = 222) | | 8 (8–12) |
| ≤40 | 198 (89) | |
| >40 | 24 (11) | |
| >80 | 4 (2) | |
| Procalcitonin (ng/mL; n = 125) | | 0.1 (0.1–0.11) |
| ≤0.25 | 109 (87) | |
| >0.25 | 16 (13) | |
| >0.5 | 12 (10) | |
| Lactate (mmol/L; n = 216) | | 1.9 (1.2–2.8) |
| >3.5 | 28 (13) | |

If antibiotics had been given only to those with hypoxia or CRP of >40 mg/L, a cutoff that is more conservative than what was used in other recent studies in SSA demonstrating the safety and efficacy of CRP-based algorithms (15, 16), treatment could have been avoided in 83% (33/40) of influenza-positive and 85% (157/185) of influenza-negative children (Fig. 3). Similarly, employing a PCT cutoff of 0.5 ng/mL (23), 88% of children in our cohort would not have received antibiotics.

**Follow-up assessment.** Study staff completed follow-up assessments with 201 (89%) participants (Fig. 1). The median time to follow-up assessment was 7 days (IQR, 7 to 8). There were no deaths or additional hospitalizations reported during the follow-up period, nor did the participants seek additional care or receive treatment from a health center, traditional healer, or drug shop. A majority (59%) of caregivers reported at least one persistent symptom, but fever had resolved for all children (Table 2). A positive test for influenza at the initial visit was associated with having any persistent symptom in bivariate analysis (risk ratio [RR], 1.6; [CI, 1.4 to 2.0]; $P < 0.001$). Controlling for age, the odds of having persistent symptoms at follow-up did not differ between those who received amoxicillin or cotrimoxazole (RR, 0.73 [CI, 0.45 to 1.19]; $P = 0.20$).

## DISCUSSION

We found that the vast majority of children presenting to a peripheral health facility in southwestern Uganda with febrile ARI had mild illness, and many did not meet clinical criteria for pneumonia, but nearly all received antibiotic treatment. These findings highlight the critical need to implement simple, point-of-care diagnostics to improve case management and decrease unnecessary antibiotic use in children with ARI. Our study demonstrates that employing rapid, point-of-care pathogen-specific and biomarker assays is feasible in a peripheral health center with basic laboratory facilities, and if incorporated into clinical decision algorithms, could substantially impact clinical management and antibiotic stewardship.

The prudent use of antibiotics is particularly important at peripheral health facilities in Uganda, where drug availability is limited and stock-outs are common. In both resource-rich and resource-limited contexts, point-of-care influenza testing has been shown to safely reduce unnecessary antibiotic prescriptions (24, 25). Furthermore, identifying influenza as a cause of ARI is important for targeting interventions such as community vaccination campaigns to prevent its spread. Our findings suggest that implementation of rapid influenza testing could be feasible and impactful at peripheral health centers in Uganda. Laboratory technicians were able to collect nasopharyngeal (NP) swabs and conduct influenza testing for all participants, and almost a fifth of children tested positive. This proportion is higher than previous estimates from Uganda's sentinel surveillance network of influenza-like illness, suggesting that influenza may be an underrecognized cause of pediatric ARI (26). Although our study began enrollment during the rainy season when influenza transmission may peak, 16 of the 40 influenza cases came during the dry season months of

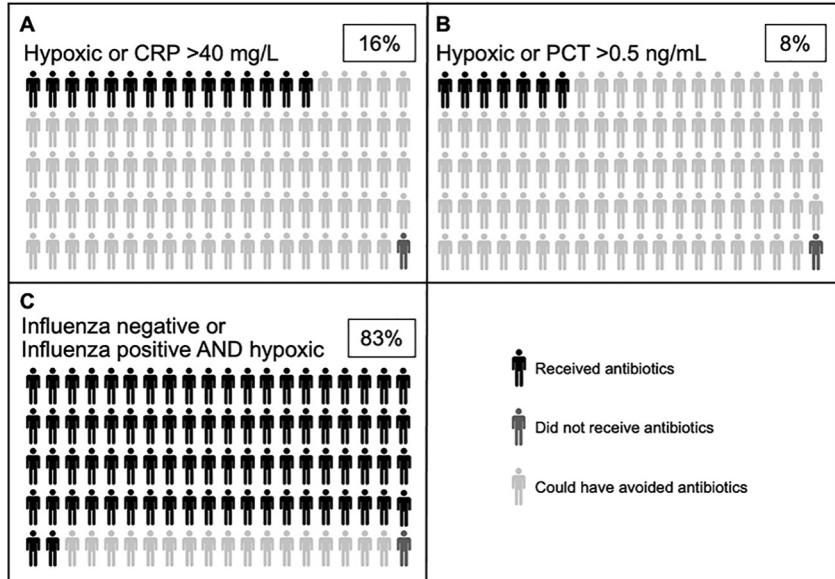

**FIG 3** Visual representation of the potential impact of point-of-care testing on antibiotic use. Dark gray figures represent the proportion of children who did not receive antibiotics in our study. The light gray figures represent the proportion of children who received antibiotics in our study but could have potentially avoided treatment if antibiotics were only given to those with (A) hypoxia (SpO2 < 90%) or CRP of >40 mg/L, (B) hypoxia or PCT of <0.5 ng/mL, or (C) negative influenza testing or positive influenza testing with hypoxia. Black figures represent children who received antibiotics in our study and would still receive them employing the additional criteria. The boxed percentages indicate the total proportion of children who would receive antibiotics if each set of criteria were employed to inform treatment decisions.

December and January, so the timing of the study alone does not fully explain the difference in estimates. The pattern of influenza cases we observed is consistent with other data from Uganda demonstrating that, in tropical climates, influenza transmission can occur year-round with seasonal peaks during the rainy seasons (27).

Clinical biomarker testing is another strategy that has been used to inform antibiotic treatment decisions (28–30). Although data regarding the ability of CRP and PCT testing to differentiate bacterial from viral or noninfectious illness are heterogeneous (31, 32), these investigations are limited by the lack of a reference standard for diagnosis of bacterial pneumonia (8, 33). Recent studies in Tanzania, Vietnam, and Thailand focusing on clinical instead of microbiological or radiographic outcomes demonstrated that leveraging clinical decision algorithms that include POC testing for CRP can reduce antibiotic use without increasing adverse outcomes among febrile children without signs of life-threatening disease in primary care settings (15, 34–36). Similar to the study conducted in Tanzania (16), few children in our cohort had elevated CRP, suggesting that implementation of this test could safely improve stewardship in our context as well. In regard to feasibility, the laboratory technicians used the NycoCard II platform to successfully measure CRP in 99% of participants. While the first 100 participants had invalid PCT results, this was due to a single technical error in the conduct of the test. Once this issue was identified and corrected with a refresher training, the laboratory staff performed PCT correctly in all of the remaining participants.

Biomarkers have also been studied in combination with pathogen-specific testing to improve stewardship and identify low-risk patients (21, 37–39). The SpUA can be positive in the setting of both colonization and infection (40), and colonization rates in SSA are high even after pneumococcal vaccination (41–43), so on its own, it is unlikely to be a useful tool for diagnosis in our target population. However, in combination with an elevated CRP, it has been shown to be sensitive and specific for primary endpoint pneumonia (21). In our cohort, none of the individuals who had a positive SpUA

had an elevated CRP, suggesting that most of these children did not have pneumococcal pneumonia and were instead colonized.

Our findings indicate that there is room for improvement in terms of adherence to national guidelines for management of ARI. Only a few children presented with danger signs, and only one had been seen previously for the same condition, indicating that the study population was representative of those with an acute, mild illness. Over half the children in our study cohort did not meet the clinical criteria for pneumonia outlined in the Ministry of Health Clinical Guidelines; guideline-concordant treatment for these children would not have included antibiotic therapy. In addition, RR was not documented by the clinician in 40% of participants, indicating that it is frequently not done as part of routine care, even though it is recommended by local guidelines. The reason for its omission is unknown; it may be skipped in well-appearing children who are breathing comfortably, or it may be considered too difficult in children who are breathing quickly and/or shallowly. In addition, previous work has shown that even when assessed, the respiratory rate is difficult to measure accurately and reproducibly (9, 10). Therefore, our findings underscore the challenges with defining pneumonia, and subsequently making antibiotic treatment decisions, based on the RR.

The strength of our study lies in its design; it was conducted in a real-world context that is representative of other peripheral health centers in resource-constrained settings, and study testing was performed by laboratory technicians already working at the clinic. However, the study also has important limitations. First, we largely selected the diagnostic tests based on test performance as reported by the manufacturer, commercial availability in Uganda, feasibility in a resource-limited context, and cost. Therefore, as we did not perform our own validation, there is the possibility of measurement error, although this would be consistent across all study participants. Second, laboratory technicians underwent formal training on the tests that were new to them prior to study start, but even with that instruction, PCT values were measured using an incorrect technique for the first 100 consecutive participants and therefore were not included in the analysis. While these missing data limit our sample size, it would not be expected to bias our results unless there was an association between the month of presentation and PCT level, which is unlikely. Third, study staff were unable to reach 10% of participants for the follow-up assessment, primarily because of lack of consistent phone access. This group may have had different outcomes than those who were able to be contacted. However, we are reassured that the group who was lost to follow-up was similar to the 90% who were reached in regard to age, home location, sex, influenza test result, CRP, and PCT. Finally, as almost all children received antibiotics, we are unable to make any conclusions related to their outcomes had antibiotics not been given in concordance with biomarker-guided algorithms. Further research is necessary to understand the drivers of antibiotic use at our study site and in this region of Uganda, but possibilities include limited training of clinical providers, lack of awareness of local guidelines, concern about "missing" a bacterial pneumonia and therefore erring on the side of providing treatment "just in case," lack of diagnostics to confirm an alternative, nonbacterial cause of symptoms, caregiver expectations, and clinical uncertainty (44, 45).

In conclusion, there is much room for improvement in regard to antibiotic stewardship in pediatric ARI in low- and middle-income countries. Better adherence to national guidelines based on clinical signs and symptoms would reduce antibiotic use, but over- and under-treatment would likely still occur relying on these methods alone. Although ideally multiplex and/or molecular testing for respiratory pathogens would be employed, this is unlikely to be feasible at peripheral health centers in SSA in the near future. Therefore, it remains important to consider and study the use of diagnostics that can be quickly and easily adopted, even if imperfect. Our findings demonstrate the preliminary feasibility and potential impact of employing relatively simple, rapid diagnostic tests to augment syndromic management as a means to facilitate

stewardship efforts in southwestern Uganda and underscore the importance of future trials implementing these tools.

## MATERIALS AND METHODS

**Study setting.** The study was conducted at Kasese Health Center (KHC), a level III, public health facility in a semi-urban setting in southwestern Uganda. KHC is located in Kasese Municipality, a town of approximately 100,000 residents (18). Clinical officers, nurses, and midwives employed by the Ministry of Health staff the health center, which houses inpatient, outpatient, and maternity wards as well as a basic laboratory. Available laboratory tests include malaria rapid antigen testing (mRDT) and blood smear microscopy, rapid HIV testing and CD4 count measurement, point-of-care hemoglobin measurement, blood glucose by portable glucometer, and urinalysis by dipstick. The OPD evaluates an average of 2,400 patients per month, approximately 30% of whom are under 5 years of age. Among this group, the most frequent diagnoses are pneumonia, malaria, and diarrhea.

**Study design.** We conducted a prospective cohort study of children with fever and acute respiratory symptoms. The primary outcomes of interest were the proportion of children who (i) received antibiotic treatment, (ii) met the clinical definition of pneumonia per Uganda Ministry of Health Clinical Guidelines (20), and/or (iii) tested positive for influenza by rapid test. As exploratory outcomes, we measured the distribution of clinical biomarkers and estimated the number of children presenting with febrile ARI who likely did not require antibiotic treatment.

**Study procedures.** Clinicians and laboratory technicians completed a full-day training related to study procedures prior to study initiation. Children aged 1 to 10 years were eligible to participate if they presented with documented (axillary temperature, $\geq$38°C) or subjective fever and at least one of the following respiratory signs: cough, oxygen saturation (SpO2) of <90%, or fast breathing (either subjective per the guardian or documented by the clinician at presentation as being >40 breaths per minute (bpm) for children younger than 5 years of age and >30 bpm if 5 years or older). We did not enroll children younger than 1 year of age given concerns regarding the total blood volume to be collected for study tests and stored samples and to ensure minimal risk to participants (46). Individuals were excluded if (i) the attendant caregiver declined to participate or no adult caregiver was present, (ii) mRDT performed as part of routine clinical evaluation for febrile illness was positive, or (iii) respiratory symptoms were present for $\geq$7 days.

We collected detailed demographic information, including caregiver occupation and materials used for home construction as a surrogate estimate of socioeconomic status. Health center staff performed clinical evaluation and treatment per local standards of care as described below (20). Children participating in the study also underwent additional testing for influenza and *Streptococcus pneumoniae* and measurement of CRP, PCT, and lactate levels. PCT measurements for the first 100 children were considered invalid due to technical errors. Results of study testing were not available to clinicians in real time and therefore did not impact clinical management. Study staff attempted to contact caregivers of all participants by phone 7 days after the initial visit to assess clinical status and document additional treatment or care-seeking.

Study procedures were approved by the University of North Carolina Institutional Review Board (18-2803), the Mbarara University of Science and Technology (MUST) Research Ethics Board (14/03-19), and the Uganda National Counsel for Science and Technology (HS 2361). Adult caregivers provided written informed consent, and children $\geq$8 years of age were also asked to provide verbal assent.

**Laboratory methods.** Trained laboratory technicians collected a venous blood sample, nasopharyngeal (NP) swabs, and a voided urine specimen from participating children. All study test kits were stored in accordance with manufacturer's instructions and utilized prior to the date of expiration. Testing was performed on site in the KHC laboratory. Specific information on the type and manufacturer of each test is shown in Table 4. All assays were conducted per manufacturer's instructions.

**Clinical definitions and local standards of care.** In routine practice at KHC, the clinical evaluation usually consists of a brief clinical history and focused physical exam. Weight, temperature, respiratory rate (RR), heart rate, and oxygen saturation are measured if equipment to do so is available. Children with fever are tested for *Plasmodium falciparum* malaria using a histidine rich protein-2 (HRP2)-based rapid antigen test (mRDT). The Ugandan national guidelines define pneumonia as the presence of cough or difficulty breathing and chest indrawing or fast breathing (RR of 50 in children aged 2 to 12 months and 40 in children aged 1 to 5 years) (20). If children meet these criteria, it is recommended that they be treated with oral amoxicillin. If the child presents with cough and fever, but no signs of pneumonia or severe disease defined by the presence of cyanosis or oxygen saturation of <90%, convulsions, inability to feed/vomiting everything, lethargy or decreased level of consciousness, or severe respiratory distress (grunting, severe chest in-drawing, nasal flaring), symptomatic care and close follow-up are advised. For the purpose of this study, children were considered to be tachypneic if the RR was greater than 40 bpm for children younger than 5 years of age and >30 bpm if 5 years or older, consistent with Ugandan national guidelines (20). Hypoxia was defined as an SpO2 of <90% as measured by a portable pulse oximeter provided by the study. Clinicians were familiar with the use and interpretation of this device, as pulse oximeters were intermittently available at the health center prior to the study start. Chest radiographs were not available at the study site and are not part of routine diagnosis of pneumonia in the majority of health facilities in Uganda.

**Statistical analysis.** Data were initially collected on paper forms by the clinicians, laboratory technicians, and study staff and then entered into a secure, Web-based Research Electronic Data Capture (REDCap) database by the study coordinator (47). Statistical analysis was performed with STATA 17

**TABLE 4** Description of diagnostic tests conducted as part of the study to evaluate malaria-negative acute respiratory illness in children presenting to the outpatient department at Kasese Health Centre from October 2019 to January 2020

| Test name | Manufacturer | Source | Assay specifications | Test performance/previous studies | Regulatory information |
|---|---|---|---|---|---|
| SD Bioline Influenza A/B Ultra | Formerly Standard Diagnostics, South Korea, now Abbott Diagnostics, USA | Local distributor | Qualitative rapid immunochromatographic cassette-based antigen assay NP swab specimen TAT[a]: 10 minutes | Per package insert, the sensitivity and specificity of the assay compared to PCR is 100% for both influenza A and B; reported sensitivity in published studies is 50–80% (48–50). | CE Mark |
| BinaxNOW *Streptococcus pneumoniae* | Abbott Diagnostics, USA | Local distributor | Qualitative rapid immunochromatographic card-based antigen assay Urine specimen TAT: 20 minutes | Limited in its specificity in children because it detects nasopharyngeal colonization in addition to pneumonia (40, 51). However, in combination with elevated C-reactive protein, a positive test has been shown to be strongly predictive of pneumococcal community-acquired pneumonia (21). | CE Mark, FDA-cleared (IVD)[b] |
| NycoCard II C-reactive protein | Abbott Diagnostics, USA | Local distributor | Battery-powered reader Whole blood, serum, plasma Analytical range for whole blood, 8–120 mg/L. TAT: 15 minutes | Has been employed in large clinical studies in low-and-middle-income settings (15, 34, 35, 52) and as the reference test for evaluation of newer diagnostics (53). | CE Mark, FDA-cleared (IVD) |
| AFIAS-1 Procalcitonin | Boditech, South Korea | Local distributor | Fluorescence-scanning instrument requiring electricity Whole blood, serum, plasma Analytical measuring range for whole blood, 0.1–100 ng/mL. TAT: 15 minutes | The package insert and one published study report high coefficients of correlation between the AFIAS PCT measurement and PCT measurements obtained using laboratory-based platforms, the Cobas e411 (Roche Diagnostics, Inc., Switzerland) and Kryptor Compact Plus (B.R.A.H.M.S. AG, Germany) (54). | CE Mark |
| Lactate Plus Lactate | Nova Biomedical, USA | Direct from manufacturer | Handheld reader Capillary or whole blood Result range 0.3–25.0 mmol/L TAT: <5 minutes | Accurate and reproducible compared to a reference analyzer (55). | None |

[a]TAT: Approximate turnaround time, including hands-on time for sample preparation and assay completion.
[b]IVD, in vitro diagnostic.

(College Station, TX). If recorded CRP and PCT measurements were below the analytical range of each test, they were replaced with the value of the lower limit of detection for the specific assay performed. Similarly, if measurements were above the analytical range, the upper limit value was assigned. We employed standard summary statistics to describe the demographic and clinical characteristics of the study cohort. Wilcoxon rank sum tests were used to compare biomarker values by (i) influenza test result and (ii) whether the child met clinical criteria for pneumonia per national guidelines and to evaluate for differences in age by (i) influenza test result and (ii) antibiotic prescribed; $P$ values are reported. The relationship between categorical demographic variables and influenza test result was analyzed using the Chi-square test or Fisher's exact test when contingency table cell counts were less than 5. Spearman's correlation coefficient was calculated to assess the strength of association between biomarkers when considered continuous variables, and Cohen's kappa statistic was calculated to assess concordance between biomarker tests using binary cutoffs. We performed log-binomial regression to assess for associations between covariates and persistence of symptoms at follow-up. $P$ values of $<0.05$ were considered statistically significant.

## ACKNOWLEDGMENTS

We sincerely thank the children and caregivers for their participation in the study. In addition, we greatly appreciate the Kasese Health Center clinicians, laboratory staff, and volunteers for their role in the conduct of the study. Finally, we thank Bonnie Shook-Sa and Melissa Miller for their advice and review of the manuscript.

This study was supported by a grant from the National Institute of Allergy and Infectious Diseases (J.J.J.; 1K24AI134990). R.M.B. is supported by the National Institutes of Health (K23AI141764) and received additional funds from a Caregivers at Carolina Award made by the Doris Duke Charitable Foundation (award 2015213). Medilink Lab & Surgicals, Ltd. (Kampala, Uganda), donated some of the influenza test kits for the study. E.J.C. received salary support through a grant from the National Heart, Lung, and Blood Institute (5T32HL007106-43). The use of REDCap was supported by the National Center for Advancing Translational Sciences (NCATS), National Institutes of Health, through grant award number UL1TR001111. The content is solely the responsibility of the authors and does not necessarily represent the official views of the NIH.

We declare no conflicts of interest.

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
