## [Reviewer comments · Microbiology Spectrum]

Microbiology Spectrum

Rapid diagnostic tests to guide case management of and improve antibiotic stewardship for pediatric acute respiratory illnesses in resource-constrained settings: a prospective cohort study in southwestern Uganda

Emily Ciccone, Lydia Kabugho, Emmanuel Baguma, Rabbison Muhindo, Jonathan Juliano, Edgar Mulogo, and Ross Boyce

Corresponding Author(s): Emily Ciccone, University of North Carolina School of Medicine

Review Timeline:

Submission Date:

October 10, 2021

Accepted:

October 19, 2021

Editor: Jennifer Dien Bard

Reviewer(s): The reviewers have opted to remain anonymous.

Transaction Report:

DOI: <https://doi.org/10.1128/Spectrum.01694-21>

October 19, 2021

Dr. Emily J Ciccone
University of North Carolina School of Medicine
111 Mason Farm Road, MBRB 2341C, CB #7036
CB #7036
Chapel Hill, NC 27599-7036

Re: Spectrum01694-21 (Rapid diagnostic tests to guide case management of and improve antibiotic stewardship for pediatric acute respiratory illnesses in resource-constrained settings: a prospective cohort study in southwestern Uganda)

Dear Dr. Emily J Ciccone:

I am please to share that your manuscript has been accepted for publication in Microbiology Spectrum, and I am forwarding it to the ASM Journals Department for publication. You will be notified when your proofs are ready to be viewed.

Sincerely,

Jennifer Dien Bard
Editor, Microbiology Spectrum
